# Epidemiology of Large Language Models: A Benchmark for Observational Distribution Knowledge

## Abstract

Artificial intelligence (AI) systems hold great promise for advancing various scientific disciplines, and are increasingly used in real-world applications. Despite their remarkable progress, further capabilities are expected in order to achieve more general types of intelligence. A critical distinction in this context is between *factual* knowledge, which can be evaluated against true or false answers (e.g., "what is the capital of England?"), and *probabilistic* knowledge, which reflects probabilistic properties of the real world (e.g., "what is the sex of a computer science graduate in the US?"). Much of previous work on evaluating large language models (LLMs) focuses on factual knowledge, while in this paper, our goal is to build a benchmark for understanding the capabilities of LLMs in terms of knowledge of probability distributions describing the real world. Given that LLMs are trained on vast amounts of text, it may be plausible that they internalize aspects of these distributions. Indeed, this idea has gained traction, with LLMs being touted as powerful and universal approximators of real-world distributions. At the same time, classical results in statistics, known under the term curse of dimensionality, highlight fundamental challenges in learning distributions in high dimensions, challenging the notion of universal distributional learning. In this work, we develop the first benchmark to directly test this hypothesis, evaluating whether LLMs have access to empirical distributions describing real-world populations across domains such as economics, health, education, and social behavior. Our results demonstrate that LLMs perform poorly overall, and do not seem to internalize real-world statistics naturally. This finding also has important implications that can be interpreted in the context of Pearl's Causal Hierarchy (PCH). Our benchmark demonstrates that language models do not contain knowledge on *observational* distributions (Layer 1 of the PCH), and thus the Causal Hierarchy Theorem implies that interventional (Layer 2) and counterfactual (Layer 3) knowledge of these models is also limited.

## 1 Introduction

Artificial intelligence (AI) systems hold great potential to accelerate many scientific disciplines including healthcare (Jiang et al., 2017; Shaheen, 2021), education (Holmes & Tuomi, 2022), and economics (Bickley et al., 2022). Automated systems based on AI are also increasingly used in a wide variety of real-world settings. Yet, despite rapid progress and broad adoption of these systems, many core capabilities required for robust and reliable AI still need to be developed further, on the path to reaching artificial human intelligence (Bubeck et al., 2023; Kaplan et al., 2020). One of the hallmark features expected from AI systems is the ability to reason probabilistically about the world, and not only to mimic human intuition and language. Such capabilities are needed for systems capable of assisting scientific discovery, advancing our understanding of complex phenomena, and help societal decision-making.

In this context, it is helpful to distinguish between two types of knowledge embedded in AI systems: *factual* knowledge, which can be evaluated based on true or false answers based on questions for which there is an agreed upon answer (e.g., "what is the capital of England?"); and *probabilistic* knowledge, which reflects the inherent uncertainties of the world we operate in (e.g., question such as "what is the sex of a computer science graduate in the US?"). Much of the current evaluations of large

language models (LLMs), currently the prevailing paradigm in AI research (Bommasani et al., 2021), has focused on factual knowledge, i.e., question answering, fact recall, etc. At the same, the degree to which LLMs encode probabilistic knowledge about the real world remains relatively underexplored.

In this work, our focus is on *observational* probabilistic knowledge, where the term observational refers to the first layer of Pearl's Causal Hierarchy (PCH, for short) Pearl (2000); Bareinboim et al. (2022). The PCH distinguishes different types of probabilistic knowledge, corresponding to different cognitive capabilities of (AI) systems: Layer 1 of *observation* (seeing), which includes reasoning about observed events and correlations between variables; Layer 2 of *intervention* (doing), which includes inferences about systems in which causal interventions are performed; and Layer 3 of *counterfactual reasoning* (imagining, see Fig. 2 left), which allows inference for hypothesized events for which the required pre-conditions may not have materialized in reality. Important results from previous work demonstrate that, in absence of appropriate assumptions, moving across different layers of PCH is provably impossible. The result known as the Causal Hierarchy Theorem (CHT, (Bareinboim et al., 2022)) states that observational knowledge (Layer 1) about a system underdetermines its interventional (Layer 2) and counterfactual (Layer 3) behavior. With causal modeling and assumptions, inferences across layers may become feasible. A classical example is causal modeling that uses observational data, combined with structural assumptions, to infer interventional or counterfactual distributions Pearl (2000); Bareinboim & Pearl (2016). Therefore, if one can show that LLMs do not have access to observational knowledge (Layer 1), based on the CHT, one may also be skeptical of these models' capabilities for reliable statements about interventions or counterfactuals.

Given the abundance of observational data, it is plausible that LLMs, trained on vast corpora of text data, have internalized aspects of real-world observational distributions. Indeed, strong claims in this debate have emerged, with OpenAI's CEO Sam Altman suggesting that "humanity has found a universal way to approximate distributions," (Altman, 2024) implying that large models serve as powerful approximators of the world, and may have access to any kind of Layer 1 knowledge.

While LLMs have been tremendously successful at various tasks, some caution is nonetheless warranted. Classical results in statistics highlight fundamental limitations of learning distributions in higher dimensions: the famous *curse of dimensionality* results demonstrate theoretically that learning distributions becomes exponentially harder (in terms of required samples) as dimensionality grows. Seminal work by Charles Stone Stone (1982) establishes that estimation rates degrade sharply with dimensionality, with the optimal rate of convergence for an unknown differentiable regression function $f$ being $\mathcal{O}(n^{-\frac{1}{2+d}})$ in any $L^p$ norm, where $d$ is the dimension of the input of $f$. Such results challenge the notion of universal distributional learning in higher dimensions, in stark contrast with the notion of LLMs working as universal approximators.

This tension raises a natural question: should one believe the optimism of universal approximation put forward by Sam Altman, or the caution grounded in statistical theory, such as the results of Charles Stone? In this paper, we attempt to shed light on this debate and examine different aspects of LLMs' capabilities in approximating distributions. We introduce a benchmark specifically designed to assess whether LLMs have access to observational distributions describing real-world populations, across various domains including economics, health, education, crime, and social behavior. This is why our approach can be termed as epidemiology, from Greek *epi* (fall upon) + *demos* (population) + *logos* (knowledge) – the knowledge of what falls upon populations.

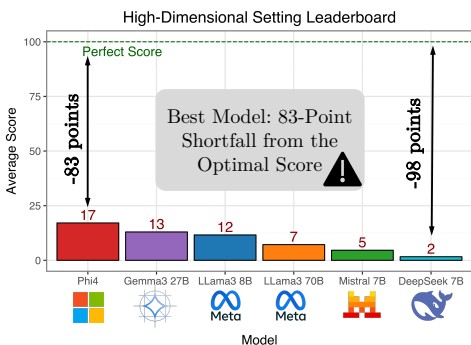

Figure 1: Benchmark leaderboard.

## 1.1 CONTRIBUTIONS

Our contribution in this work is to construct a benchmark for evaluating the capabilities of large language models in terms of access to knowledge about observational distributions in the real-world.

(i) We curate a total of 10 datasets, ranging from healthcare, health behavior, and education, to labor, consumer spending, and crime statistics. For each large scale dataset, which describes

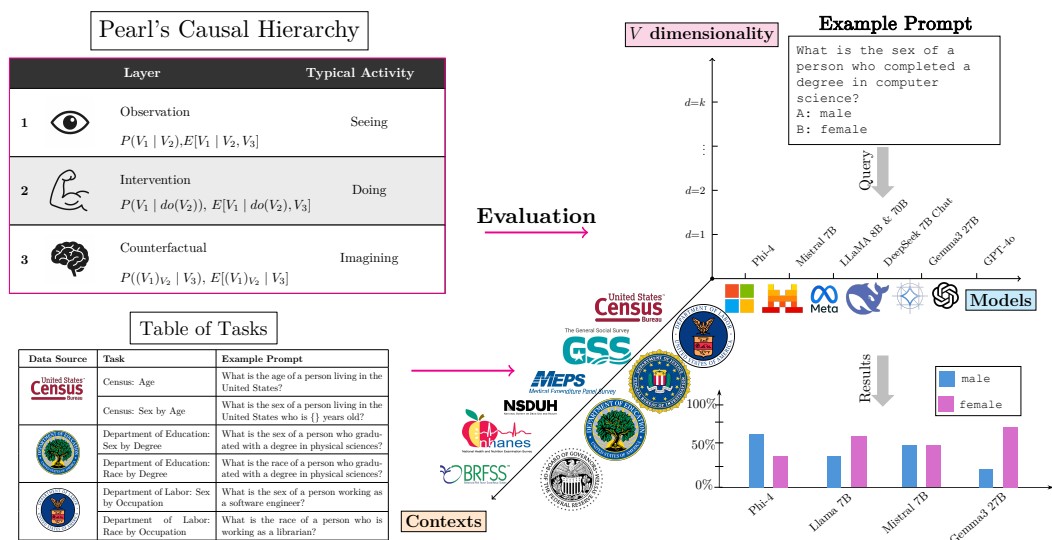

Figure 2: Overview figure for our benchmark.

the population level statistics in the United States, we extract a set of questions to test whether a language model has knowledge about the population encoded in the respective dataset.

(ii) We design a modular evaluation framework that facilitates easy benchmarking of additional models, making it straightforward to add datasets, questions, or models to our benchmark.

A preview of our results is shown in Fig. 1, indicating that over a range of models, the performance of the current generation of LLMs in terms of knowledge of real-world distributions is quite poor. As we argue later in the text, access to observational distributions is a fundamental step towards performing interventional (Layer 2) or counterfactual (Layer 3) inference, implying that are our results also implicitly challenge the abilities of LLMs in higher layers of Pearl's Hierarchy.

**Related Literature.** A growing body of literature investigates aspects of knowledge and generalization abilities of LLMs. The two works most related to ours are that of Santurkar et al. (2023), which examines the responses on language models on Pew Research Center's American Trends Panel (Pew Research Center, 2024), in order to investigate if LLMs exhibit responses of a specific demographic group. Another related work is that of Dominguez-Olmedo et al. (2024), that examines the responses of language models on the American Community Survey (ACS) (U.S. Census Bureau, 2024), and compares them to the real responses of individual. Our work takes a broader view, and attempts to build a benchmark for systematically analyzing the knowledge of models on observational distributions (Layer 1 of the PCH), across a wide range of datasets, and distributions over variables with different dimensions.

Other works investigate the ability of models to perform correct probabilistic inferences, given information in forms of probability tables or probability conditionals (He-Yueya et al., 2023; Jin et al., 2023; Nafar et al., 2024). This line of work is distinct from ours, since our benchmark aims to evaluate the availability of the true probabilities to the model, rather than the validity of inferences over these probabilities. We also mention the work of Zhao et al. (2023), aimed at testing factual correctness of model responses on various domains, including knowledge about the world, math, or reasoning. In parallel, other lines of work use the term faithfulness while evaluating language models, usually referring to faithfulness of model responses to the the actual source material where the information is available. For instance, (Zhou et al., 2023) investigates whether responses of models are affected by knowledge conflict, and whether they exhibit appropriate response abstentions. This work is focused on testing factual knowledge, as opposed to descriptive knowledge investigated by our benchmark. Another body of work looks at faithfulness of model explanations, that is, whether the explanation cited by the model is in fact compatible with the model's actual reasoning steps (Agarwal et al., 2024; Matton et al., 2025), differing from the approach and the goal of our work.

## 2 BENCHMARK AXES

We use the language of structural causal models (SCMs) (Pearl, 2000). An SCM is a tuple $\mathcal{M} := \langle V, U, \mathcal{F}, P(u) \rangle$, where $V, U$ are sets of endogenous (observable) and exogenous (latent) variables, respectively, $\mathcal{F}$ is a set of functions $f_{V_i}$, one for each $V_i \in V$, where $V_i \leftarrow f_{V_i}(\text{pa}(V_i), U_{V_i})$ for some $\text{pa}(V_i) \subseteq V$ and $U_{V_i} \subseteq U$. The assignment mechanisms $\mathcal{F}$ determine how each variable attains its value, and the set $\text{pa}(V_i)$ is called the parent set of $V_i$. Together with the probability distribution $P(u)$ over the exogenous variables $U$, the SCM specifies the entire behavior of the underlying phenomenon, meaning that it fully specifies all observational, interventional, and counterfactual distributions (Bareinboim et al., 2022).

The remainder of the section is organized according to Fig. 2, and the different axes appearing therein. We first discuss observational knowledge and its different dimensionalities (Sec. 2.1), followed by the description of datasets that are used (Sec. 2.2), and then the tasks constructed for eliciting models' capabilities (Sec. 2.2.1). In Sec. 2.3, we discuss different models currently included in the benchmark.

### 2.1 AXIS 1: DIMENSIONALITY OF LAYER 1 (OBSERVATIONAL) KNOWLEDGE

Our focus in this paper is on evaluating observational (Layer 1) knowlege available in AI systems. The *observational distribution* of the underlying phenomenon can be defined through the SCM itself:

**Definition 1** (Observational Distribution (Bareinboim et al., 2022)). *An SCM $\mathcal{M}$ that is a 4-tuple $\langle V, U, \mathcal{F}, P(u) \rangle$ induces a joint probability distribution $P(V)$ such that for each $Y \subseteq V$,*

$$P^{\mathcal{M}}(Y = y) = \sum_u \mathbb{1}\Big(Y(u) = y\Big) P(U = u), \tag{1}$$

*where $Y(u)$ is the solution for $Y$ after evaluating $\mathcal{F}$ with $U = u$.*

In the above definition, $Y \subseteq V$ should be thought of as multidimensional. A fixed value of exogenous $U = u$ is called a *unit*, and can be thought of as a individual in the population. To obtain the observational distribution $P(Y = y)$, we go over all units $u$, and add up the probability mass $P(U = u)$ of all units for which $Y$ attains the value $y$. The observational distribution $P(V)$ captures all probabilistic relationships among observed variables under passive observation. On top of the distribution, we can also consider other types of quantities, such as (conditional) moments. In particular, we may consider different types of knowledge, such as:

(i) **Marginal expectations:** e.g., $\mathbb{E}[V_1]$, the average value of a single variable.

(ii) **Marginal distributions:** e.g., $P(V_1)$, the full distribution of a variable.

(iii) **Conditional expectations:** e.g., $\mathbb{E}[V_1|V_2]$, the average value conditioned on another variable.

(iv) **Conditional distributions:** e.g., $P(V_1|V_2)$, the distribution of one variable given another.

(v) **Joint distributions:** e.g., $P(V_1, \ldots, V_k)$, the distribution of over the variables $V_1, \ldots, V_k$.

The above types of knowledge can be viewed as a hierarchy, with the level of required knowledge being progressively more refined. Our goal can be described as follows: we design tasks probing the above types of knowledge, based on large-scale datasets describing populations. For instance, we refer to a dataset $\mathcal{D}$, and consider the distribution over covariates $P(V)$ implied by $\mathcal{D}$. Then, we elicit responses from an LLM about the same distribution. The model's distribution is labeled $\tilde{P}(V)$. Our goal is to then compare various aspects of $\tilde{P}(V)$ and $P(V)$, according to types of knowledge (i)-(v). In this way, we can systematically evaluate whether LLMs internalize such quantities over real-world populations. In fact, we can formally argue that testing knowledge on observational distributions is a critical first step for understanding knowledge on causal capabilities more broadly:

**Proposition 1** (Causal Hierarchy Theorem (Bareinboim et al., 2022)). *Let $\mathcal{M}$ be an SCM, $P(V)$ be its observational distribution, and $\mathcal{A}$ a set of causal assumptions encoded in the form of a causal diagram (Bareinboim et al., 2022) or ignorability statements (Rubin, 1974; Pearl, 2000). Then,*

*(a) in absence of $\mathcal{A}$, $P(V)$ underdetermines interventional and counterfactual distributions,*

*(b) in absence of $P(V)$, $\mathcal{A}$ underdetermines interventional and counterfactual distributions.*

The first part of the above proposition is the one commonly considered in the causal inference literature – in short, it states that, in absence of causal assumptions, it is generally impossible to provide any guarantees for inference over interventional or counterfactual distributions. The second part states that in absence of the observational distribution, no inferences can be made for higher layers of the PCH, leading to the following important corollary:

**Corollary 2** (No Observational Distribution $\implies$ No Layer 2/3 Inference). *If a model's distribution $\tilde{P}(V)$ differs from the true $P(V)$, no guarantees can be provided for the validity of the model's interventional or counterfactual inferences.*

The above corollary captures an important motivation for the benchmark constructed in this paper – even if we assume that an LLM has access to correct causal knowledge (a point we do not investigate in this manuscript), it still may not be able to perform causal inferences in case its observational distribution does not match the one in the real world. Therefore, testing whether language models have access to observational knowledge (required for causal reasoning) is an important first step since a negative answer warrants skepticism about the model's capabilities for performing claims in higher layers of the PCH.

## 2.2 Axis 2: Datasets Describing Populations

As mentioned in the previous section, we will leverage large-scale datasets that describe population level statistics in order to establish the ground truth observational distribution $P(V)$. After this, we will probe the language model's distribution $\tilde{P}(V)$ to draw comparisons. In this work, for the observational distributions $P(V)$, we make use of ten large, publicly available datasets that collectively describe diverse aspects of the US population, and are often considered representative of the national level statistics:

(1) **American Community Survey (ACS) 2023** (U.S. Census Bureau, 2023): Conducted by the US Census Bureau, the ACS collects detailed demographic, social, economic, and housing data annually. Our focus is on income, education, and employment information across demographics.

(2) **National Health and Nutrition Examination Survey (NHANES) 2021-2023** (National Center for Health Statistics, 2023): Administered by the Centers for Disease Control and Prevention (CDC), NHANES combines interviews and physical exams to assess health and nutrition of individuals in the US. We investigate obesity, diabetes, and dietary habits across demographics.

(3) **Behavioral Risk Factor Surveillance System (BRFSS) 2023** (Centers for Disease Control and Prevention, 2023): A telephone survey system run by the CDC, tracking health-related risk behaviors and conditions. We investigate exercise habits, diabetes, blood pressure, asthma, cholesterol, visual/auditive impairments by US states.

(4) **Medical Expenditure Panel Survey (MEPS) 2023** (Agency for Healthcare Research and Quality, 2023): A set of surveys conducted by the Agency for Healthcare Research and Quality (AHRQ), measuring health services use, expenditures, and insurance coverage. We investigate healthcare expenditure, utilization, and insurance across demographics.

(5) **National Survey on Drug Use and Health (NSDUH) 2023** (Substance Abuse and Mental Health Services Administration, 2023): Collected by Substance Abuse and Mental Health Services Administration (SAMHSA), the NSDUH provides information on substance use and mental health in the US. We investigate alcohol and drug use across demographics.

(6) **Survey of Consumer Finances (SCF) 2022** (Board of Governors of the Federal Reserve System, 2022): Sponsored by the Federal Reserve Board, the SCF provides detailed data on US household finances. We analyze food expenditure, home ownership, assets, and debt across demographics.

(7) **General Social Survey (GSS) 2022** (NORC at the University of Chicago, 2022): Conducted by the National Opinion Research Center (NORC) at the University of Chicago, the GSS collects data on social attitudes, behaviors, and demographics of US adults. We investigate political views and party identification across age, sex, race, education, and income.

(8) **Department of Education (IPEDS)** (U.S. Department of Education, National Center for Education Statistics, 2023): The Integrated Postsecondary Education Data System (IPEDS) collects data from colleges, universities, and technical schools. We investigate college degrees by sex.

(9) **Department of Labor BLS Data 2023** (U.S. Bureau of Labor Statistics, 2023): The Bureau of Labor Statistics of the US Department of Labor provides detailed information about occupations, including worker demographics. We investigate occupations by sex and race.

(10) **Federal Bureau of Investigation (FBI) Arrest Statistics (UCR Program):** Compiled by the FBI's Uniform Crime Reporting Program (UCR), this dataset contains arrest statistics across the US. We investigate crime rates by race and sex.

Each of the datasets either reports national-level statistics, or provides data on individuals. When individual data is available, sample weights are provided (determining how many persons in the overall US population the individual represents), allowing one to obtain representative results at the national level. Together, these datasets allow for a broad evaluation of models' abilities to approximate population-level observational distributions across social, economic, educational, health, and behavioral domains. We next describe how different tasks in our benchmark are constructed.

### 2.2.1 AXIS 2 TASKS: RESPONSES AND SCORING

Our benchmark consists of different *tasks*, and each task is associated with a dataset. The task is defined by the pair $V_Y, V_X$, representing the conditional distribution $P(V_Y \mid V_X)$ (here, $V_X$ could be possibly empty). Each task is also accompanied by a natural language prompt template $\pi$ (which takes a value of $V_X = v_X$ and returns a natural language question), and a set of valid answers in the domain of $V_Y$, labeled $\mathrm{dom}(V_Y)$.

**Example 1** (NSDUH: Marijuana Usage by Age). *The NSDUH dataset tracks various aspects of addiction and mental health for the US population. Suppose that $V_Y$ represents whether a person ever used marijuana, and $V_X$ represents age. In the benchmark, we are interested whether the LLM has access to the conditional distribution $P(V_Y \mid V_X)$. To test this, the accompanying prompt template $\pi$ for the task is given by:*

$$\pi(v_X) = \text{``Has a person aged \{v_X\} ever used marijuana?''} \tag{2}$$

*Once a specified value $v_X = 16$ is chosen, this results in a prompt "Has a person aged 16 ever used marijuana?", which is given to the model. In the prompt, the model is provided with a set of possible answers, which in this case amounts to $\mathrm{dom}(V_Y) = \{no, yes\}$. For open-weights models, we can inspect the probabilities associated with each of the responses in the model's next-token prediction. Alternatively, for closed models, the same question can be repeated multiple times, and we record all the provided answers, allowing us to reconstruct the distribution over answers. Finally, we compare the model's distribution with the true $P(V_Y \mid V_X)$ distribution from the NSDUH dataset.* □

Intuitively, the difficulty of the task should increase with larger dimensions of $V_Y$ and $V_X$. For instance, if $|V_Y| = 1, |V_X| = 0$, our task is to recover the 1-dimensional marginal distribution $P(V_Y)$. If $|V_Y| = 1, |V_X| > 0$, we are trying to recover a 1-dimensional conditional distribution $P(V_Y \mid V_X)$. The question templates $\pi$ are carefully designed, to reflect the original questions asked to individuals during data acquisition. In total, we constructed 75 tasks across 10 datasets for $|V_Y| = 1, |V_X| = 1$ (we refer to this as the low-dimensional setting), and 94 tasks across 4 datasets for $|V_Y| = 1, |V_X| \in \{2, 3, 4, 5\}$ and $V_Y$ binary (referred to as the high-dimensional setting). Fig. 2 (bottom left) offers some further examples, in addition to Ex. 1, while Appendix A contains the full list of tasks. We remark here that an alternative way of eliciting the model's knowledge also exists, captured by the following example:

**Example 1** (continued – NSDUH: Marijuana Usage by Age). *For the task of eliciting the distribution of marijuana usage by age group, a template prompt*

$$\pi^{prob}(v_X) = \text{``What is the probability that a person aged \{v_X\} ever used marijuana?''} \tag{3}$$

*may be used instead, representing an alternative way of eliciting $\tilde{P}(V_Y \mid V_X)$.* □

The comparison of the two prompting approaches (probabilistic vs. Q&A) is discussed in Appendix E, while in the main text we focus on Q&A prompt templates as in Eq. 2.

**Eliciting Responses.** As mentioned, for each question the model is asked, there is a specific range of possible responses. Our approach to obtain the model answers is to label each response in $\mathrm{dom}(V_Y)$ with labels $A, B, C, \ldots$, and then inspect the conditional probability associated with tokens $A, B, C,$

etc. Let $\mathcal{A}(v_Y)$ denote the letter of the answer $v_Y$. The model's probability associated with each answer $v_Y \in \text{dom}(V_Y)$ corresponding to label $\mathcal{A}(v_Y)$ in the prompt is then simply computed as

$$\tilde{P}(V_Y = v_Y \mid V_X = v_X) \triangleq \frac{\tilde{P}^{\text{nt}}(\text{answer } \mathcal{A}(v_Y))}{\sum_{\tilde{v}_Y \in \text{dom}(V_Y)} \tilde{P}^{\text{nt}}(\text{answer } \mathcal{A}(\tilde{v}_Y))}, \tag{4}$$

where $\tilde{P}^{\text{nt}}$ is the model's next-token probability function given the prompt.

**Example 1** (continued – NSDUH: Marijuana Usage by Age). *Suppose that following the prompt*

```
Has a person aged 16 years ever used marijuana? A. yes B. no
```

*we find the model's next-token probabilities to be $\tilde{P}^{\text{nt}}(\text{answer } A) = 0.01$, $\tilde{P}^{\text{nt}}(\text{answer } B) = 0.03$. Then, the model's probability is computed as $\tilde{P}(V_Y = 1 \mid V_X = 16) = \frac{0.01}{0.01 + 0.03} = 25\%$.* □

This is a known strategy for eliciting model responses, used in numerous works on question answering (Dominguez-Olmedo et al., 2024; Hendrycks et al., 2020; Santurkar et al., 2023). To mitigate ordering bias (Dominguez-Olmedo et al., 2024), we average over different answer permutations. For open-source models we use next-token probabilities; for closed models we use Monte Carlo bin-counting. The ground truth observational distribution $P(V_Y \mid V_X)$ is estimated based on the available dataset, using bin-counting in low-dimensional settings and `lightgbm` plus cross-validation in high-dimensional settings. See Appendix D for more details on eliciting responses.

**Scoring Strategy.** We next develop a scoring strategy for each task based on the true distribution $P(V_Y \mid V_X)$ and the model's distribution $\tilde{P}(V_Y \mid V_X)$. For this purpose, we use the $L_1$-norm (instead of KL divergence, to avoid sensitivity to low-probability events and support mismatch (Gibbs & Su, 2002)), and define the following distributional distance:

$$D(\tilde{P}||P; V_Y, V_X) = \sum_{v_X} \sum_{v_Y} \left| P(V_Y = v_Y \mid V_X = v_X) - \tilde{P}(V_Y = v_Y \mid V_X = v_X) \right| P(V_X = v_X) \tag{5}$$

The above notion of distance allows the evaluation of how far $P, \tilde{P}$ are for the specific conditional $V_Y \mid V_X$. To obtain a normalized score, we compare the distance $D(P||\tilde{P})$ to the distance of $P$ from a uniform distribution over the answers, labeled $P^{\text{unif}}$, defined as

$$P^{\text{unif}}(V_Y = v_Y \mid V_X = v_X) \triangleq \frac{1}{|\text{dom}(V_Y)|} \; \forall v_X, v_Y. \tag{6}$$

In this context, we refer to the $P^{\text{unif}}$ distribution as a random *baseline*. In the high-dimensional case, which focuses on binary $V_Y$, we add another baseline, namely a fixed 0/1 prediction depending on the dataset mean, given by $P^{0/1}(V_Y = 1 \mid V_X = v_X) \triangleq \mathbb{1}(\mathbb{E}_P[V_Y] > 0.5)$ for all $v_Y, v_X$, where $\mathbb{E}_P[V_Y]$ represent the true distribution mean. Therefore, the 0/1 baseline predicts a constant probability of 0 for all conditioning sets for outcomes with a marginal incidence $\mathbb{E}_P[V_Y] \leq 0.5$, and predicts a constant 1 whenever the marginal incidence is $\mathbb{E}_P[V_Y] > 0.5$. Such a baseline also corresponds to effectively no probabilistic knowledge. With these baselines in place, our scoring system is described next (illustrated in Fig. 3a). As many datasets consist of samples of the population (and do not survey the entire population), there is some uncertainty on the ground truth distribution $P$. In the scoring, we account for this fact, and proceed as follows. We draw bootstrap samples of size $|\mathcal{D}|$ (size of the dataset) labeled $\mathcal{D}^{(b)}$, and extract the bootstrap ground truth distribution $P^{(b)}$. We then look at the distance $D(P||P^{(b)})$ of $P$ (based on the available data) and $P^{(b)}$ (obtained from bootstrapped data), across different samples (see green density in Fig. 3a). The upper 5% quantile of the spread of $D(P||P^{(b)})$ is labeled with the perfect score $S = 100$ – meaning that the score 100 is assigned if a model's distribution cannot be statistically distinguished from the ground truth at the 5% significance level. For setting the score of 0, we use the minimum of the distances $D(P||P^{\text{unif}})$, $D(P||P^{0/1})$, with the latter only considered for binary outcomes. The model's final score is then linearly interpolated between $S = 0$ and $S = 100$, as follows:

$$S_T = 100 \times \max\left(1 - \frac{D(\tilde{P}||P)}{\min\{D(P^{\text{unif}}||P), D(P^{0,1}||P)\}}, 0\right). \tag{7}$$

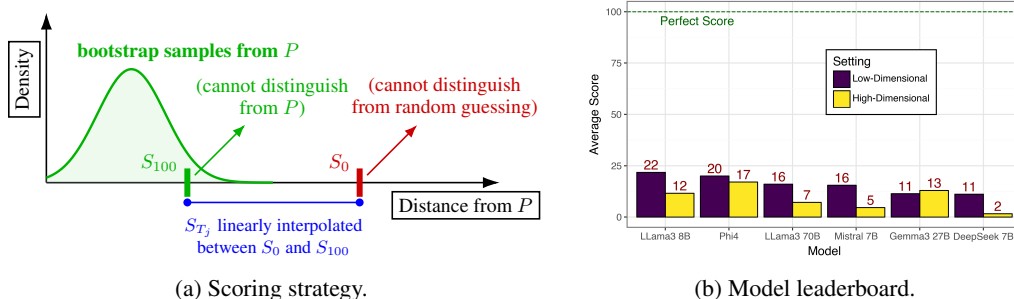

(a) Scoring strategy.
(b) Model leaderboard.

Figure 3: Scoring strategy and model performance across task groups.

## 2.3 Axis 3: Models

Models we consider can be divided in two groups: *open-weight models*, whose trained parameters are publicly available, and *closed-weight models*, whose parameters remain proprietary and can be accessed only via an API. We focus mainly on open models, but also consider some closed models to ensure our benchmark covers a broad range of state-of-the-art models. While the benchmark evaluates zero-shot performance of models, a natural question is whether fine-tuning can help boost performance. Impact of fine-tuning is investigated in Appendix F.

**Open-weight Models.** Our open-weight models include both the original pretrained checkpoints, and *instruction-tuned* variants, which are further trained on human-generated instruction–response pairs to improve their ability to follow instructions in user prompts. Particularly, we evaluate the following models: Mistral 7B (Jiang et al., 2023), LLaMA3 8B (Grattafiori et al., 2024), LLaMA3 70B (Grattafiori et al., 2024), Gemma3 27B (Team et al., 2024), DeepSeek 7B (Bi et al., 2024), Phi-4 (Abdin et al., 2024), and DeepSeek R1 32B (Bi et al., 2024). In the main text, we focus on instruction-tuned models, while Appendix B compares instruction-tuned and pretrained models.

**Closed-weight Models.** Closed-weight models are available only through API-based access to proprietary weights, and include a state-of-the-art reasoning model. In particular, we evaluate Reasoning Language Models (RLMs) such as OpenAI's o4-mini. We also include models with web access, such as OpenAI's GPT-4.1. Appendix C analyzes the performance of selected closed models.

## 2.4 Benchmark Construction

Our benchmark is designed with reproducibility and extensibility as key principles. All dataset constructions are made available, ensuring transparency and verifiability. The modular structure of the benchmark allows easy addition of both new datasets and new models, facilitating broader future evaluations. Furthermore, model evaluation is streamlined: any model available through Hugging Face (Wolf et al., 2019) can be benchmarked with a single line of code, allowing for rapid inclusion of diverse architectures. At the time of submission, our benchmark consists of 10 pre-processed datasets, 169 tasks (75 low-dimensional and 94 high-dimensional), with 12 open-weight models evaluated.

## 3 Key Insights and Observations

Results summarizing the leaderboards for the low- and high-dimensional settings are shown in Fig. 3b. They indicate that the observational distributions $\tilde{P}(V)$ encoded in the LLMs are closer to the ground truth, real-world observational distributions $P(V)$ than the uniform distribution $P^{\text{unif}}(V)$ since models achieve scores above zero. All models achieve scores better than uniform guessing, suggesting some access to L1 knowledge. However, overall, the models' performance is rather poor, with the best models scoring an average of 22/100 points in the low-dimensional, and 17/100 in the high-dimensional setting, respectively. Therefore, capabilities of LLMs for encoding real-world observational distributions seem limited, and using such knowledge for downstream tasks should be done with care. Furthermore, Appendix F shows that fine-tuning may offer limited added benefit, raising interesting questions on how to design approaches for improving the models' L1 knowledge.

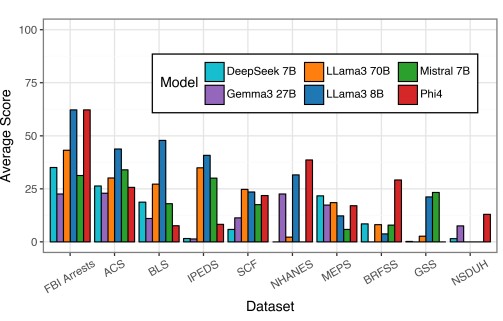
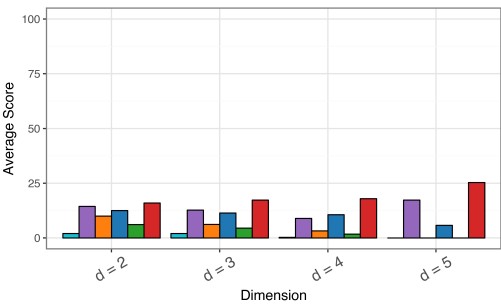

(a) Performance by dataset (low-dimensional).

(b) Performance by dimension.

Figure 4: Model performance across (a) datasets; (b) dimension.

**Low-Dimensional Setting & Performance by Dataset.** The performance of models on 75 low-dimensional tasks, comparing $\tilde{P}(V_Y \mid V_X)$ with the ground truth $P(V_Y \mid V_X)$ for $|V_Y| = |V_X| = 1$ is shown in Fig. 3b (purple bars). Among a selection of models from different organizations (Meta, Microsoft, Google, DeepSeek AI, Mistral AI), the overall performance across models does not exhibit large differences, and all models perform poorly. Another interesting aspect is the performance of models across different datasets, visualized in Fig. 4a. The best performance is observed over the FBI Arrest Statistics (sex/race distributions over crime types), ACS Census data (questions on employment, education, and income across demographics), and Departments of Education (sex by graduation degree) and Labor (sex/race by occupation). For three of these four datasets (FBI, BLS, IPEDS) the detailed statistics we queried are available on their websites, meaning that models could have had access to the exact probability tables used for constructing our questions. Therefore, improved performance may correspond to the actual data being available in the model's training set. This observation also emphasizes the importance of the high-dimensional setting – in which we condition on multiple observed variables – as this knowledge is not readily available on the web.

**High-Dimensional Setting & Performance by Dimension.** The performance of models in the high-dimensional case, for $P(V_Y \mid V_X)$ with $|V_Y| = 1, |V_X| \in \{2, 3, 4, 5\}$, and $V_Y$ binary, is shown in Fig. 3b (yellow bars). As the scores indicate, performance is again quite poor across models, and certainly worse than in the low-dimensional case. We also inspect the model performance with varying dimension (Fig. 4b). The pattern of the performances does not clearly indicate the curse of dimensionality, possibly due to the fact that each dimension $d$ has a relatively small number of tasks, and additional tasks may need to be added to investigate this effect in more detail.

**Implications on Causal Inferences.** Recovery of high-dimensional conditional distributions investigated in this work may be seen as related to questions of causal inference. Commonly, to adjust for confounding by a set of covariates $V_Z$, data analysts compute the propensity of a treatment $V_X$, that is, reweigh the samples with $P(V_X = v_X \mid V_Z = v_Z)^{-1}$ to obtain estimates of a desired potential outcome (Rubin, 1977; Rosenbaum & Rubin, 1983). Other strategies use a predictive model for the outcome for the outcome distribution $P(V_Y \mid V_X, V_Z)$ in order to allow for or improve causal inference (Bang & Robins, 2005). However, if such conditional distributions are not available to a model, eliciting them may be a futile exercise. Therefore, our work brings into question some recent approaches claiming to improve causal inferences using foundation models (De Bartolomeis et al., 2025). In our view, using AI models for causal inference may require a more sophisticated approach.

**Limitations & Future Work.** A limited number of models are evaluated on our benchmark. However, by choosing a representative sample of models, we cover a range of models developed by different organizations. Furthermore, our emphasis on easy access and evaluation of models will facilitate the benchmarking of other models in future work. Another limitation is that high-dimensional inference is investigated across 4 datasets, and the number of contexts for evaluation should grow over time. We further envision that our work will set the basis for evaluating capabilities of AI models for inference in Layers 2 & 3 of the PCH – to systematically test how observational distributions available to models affect downstream tasks such as causal or counterfactual inferences.

**Ethics statement.** This work introduces a benchmark based on publicly available datasets and models, with all assets cited and used in accordance with their licenses. No new data collection, human subjects, or sensitive information are involved. We do not foresee ethical risks such as privacy violations, bias amplification, or harmful applications beyond those already present in the underlying datasets and models. In fact, our benchmark may contribute positively by encouraging researchers to critically evaluate model capabilities for observational knowledge before deployment in downstream applications such as causal inference.

**Reproducibility statement.** Strong emphasis is placed on reproducibility. All datasets used are publicly accessible, and we release preprocessing scripts, evaluation code, and instructions to reproduce every reported result. The benchmark is designed to make reproducing results and evaluating new models straightforward. We provide all experimental details, including data splits, hyperparameters, evaluation metrics, and compute resources. Code and documentation are made available in the anonymized code repository `https://anonymous.4open.science/r/llm-epidemia-563D`.

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

# SUPPLEMENTARY MATERIAL FOR *Epidemiology of Large Language Models: A Benchmark for Observational Distribution Knowledge*

The source code for reproducing the benchmark can be found in the anonymized code repository `https://anonymous.4open.science/r/llm-epidemia-563D`. The repository includes a README file explaining all the steps for setting up the benchmark. For the Llama3 8B, Mistral 7B, and Phi4 models the experiments were run on a single NVIDIA H100 GPU. Evaluating both the low- and high-dimensional settings required <1 hour per model. Evaluating DeepSeek 7B (Chat version), Gemma3 27B, and LLama3 70B models was done on a node with four NVIDIA GH200 Grace Hopper chips, and running both settings required <1 hour of compute per model.

## A  TASKS

This appendix contains the full list of tasks considered in the benchmark, complementing the description of task construction from Sec. 2.2.1. In Tab. 1, the 75 tasks for the low-dimensional setting are listed, which are concerned with recovering $P(V_Y \mid V_X)$ for univariate $V_X, V_Y$. Tab. 2 lists all the 94 tasks for the high-dimensional setting, recovering $P(V_Y \mid V_X)$ for $V_Y$ univariate and binary, and $|V_X| \in \{2, 3, 4, 5\}$.

Table 1: Low-dimensional tasks.

| Task # | Task Name |
|---|---|
| 1 | Census: Employment Status by Sex |
| 2 | Census: Employment Status by Race |
| 3 | Census: Employment Status by Age |
| 4 | Census: Employer by Sex |
| 5 | Census: Employer by Race |
| 6 | Census: Employer by Age |
| 7 | Census: Salary by Sex |
| 8 | Census: Salary by Race |
| 9 | Census: Salary by Age |
| 10 | Census: Education by Sex |
| 11 | Census: Education by Race |
| 12 | Census: Education by Age |
| 13 | BRFSS: Exercise by State |
| 14 | BRFSS: Diabetes by State |
| 15 | BRFSS: High BP by State |
| 16 | BRFSS: Asthma by State |
| 17 | BRFSS: Cholesterol by State |
| 18 | BRFSS: Visual Impairments by State |
| 19 | BRFSS: Hearing Impairments by State |
| 20 | BRFSS: Heart Attack by State |
| 21 | BRFSS: Stroke by State |
| 22 | Department of Education: Sex by Type of Degree |
| 23 | FBI Crime Statistics: Sex by Crime Type |
| 24 | FBI Crime Statistics: Race by Crime Type |
| 25 | GSS: Political View by Age |
| 26 | GSS: Political View by Race |
| 27 | GSS: Political View by Education |
| 28 | GSS: Political View by Income |
| 29 | GSS: Political View by Sex |
| 30 | GSS: Party Affiliation by Age |
| 31 | GSS: Party Affiliation by Race |
| 32 | GSS: Party Affiliation by Education |
| 33 | GSS: Party Affiliation by Income |
| 34 | GSS: Party Affiliation by Sex |
| 35 | Department of Labor: Sex by Occupation |

| Task # | Task Name |
|---|---|
| 36 | Department of Labor: Race by Occupation |
| 37 | MEPS: Expenditure by Age Group |
| 38 | MEPS: Office-based Visits by Age Group |
| 39 | MEPS: Inpatient Visits by Age Group |
| 40 | MEPS: Dental Visits by Age Group |
| 41 | MEPS: Has Insurance by Age Group |
| 42 | MEPS: Expenditure by Race |
| 43 | MEPS: Office-based Visits by Race |
| 44 | MEPS: Inpatient Visits by Race |
| 45 | MEPS: Dental Visits by Race |
| 46 | MEPS: Has Insurance by Race |
| 47 | NHANES: Age by BMI Group |
| 48 | NHANES: Diabetes by BMI Group |
| 49 | NHANES: Diabetes by Age Group |
| 50 | NHANES: Weekly Alcohol Consumption by Age Group |
| 51 | NSDUH: Alcohol Use in Last Month by Age |
| 52 | NSDUH: Cigarette Use in Last Month by Age |
| 53 | NSDUH: Marijuana Ever Used by Age |
| 54 | NSDUH: Cocaine Ever Used by Age |
| 55 | NSDUH: Heroin Ever Used by Age |
| 56 | NSDUH: Alcohol Use in Last Month by Race |
| 57 | NSDUH: Cigarette Use in Last Month by Race |
| 58 | NSDUH: Marijuana Ever Used by Race |
| 59 | NSDUH: Cocaine Ever Used by Race |
| 60 | NSDUH: Heroin Ever Used by Race |
| 61 | SCF: Food Expenditure by Age Group |
| 62 | SCF: House Ownership by Age Group |
| 63 | SCF: Total Assets by Age Group |
| 64 | SCF: Debt by Age Group |
| 65 | SCF: Net Worth by Age Group |
| 66 | SCF: Food Expenditure by Race |
| 67 | SCF: House Ownership by Race |
| 68 | SCF: Total Assets by Race |
| 69 | SCF: Debt by Race |
| 70 | SCF: Net Worth by Race |
| 71 | SCF: Food Expenditure by Education |
| 72 | SCF: House Ownership by Education |
| 73 | SCF: Total Assets by Education |
| 74 | SCF: Debt by Education |
| 75 | SCF: Net Worth by Education |

Table 2: High-dimensional tasks. The third column indicates the task dimension $d = |V_X|$, and the fourth column indicates whether the task is used for the evaluation of closed models in Appendix C.

| Task # | Task Name | $d$ | Closed Eval |
|---|---|---|---|
| 1 | BRFSS: Diabetes by Sex, Race | 2 | ✔ |
| 2 | BRFSS: Diabetes by Age, Income | 2 | ✔ |
| 3 | BRFSS: Diabetes by Age, Race | 2 | ✔ |
| 4 | BRFSS: Diabetes by Sex, Income | 2 | ✔ |
| 5 | BRFSS: Diabetes by Age, Sex, Race | 3 | ✔ |
| 6 | BRFSS: Diabetes by Age, Race, Income | 3 | ✘ |
| 7 | BRFSS: Diabetes by Age, Education, Income | 3 | ✔ |
| 8 | BRFSS: Diabetes by Age, Sex, Income | 3 | ✔ |
| 9 | BRFSS: Diabetes by Age, Education, Sex, Race | 4 | ✘ |
| 10 | BRFSS: Diabetes by Age, Sex, Race, Income | 4 | ✘ |
| 11 | BRFSS: Diabetes by Age, Education, Race, Income | 4 | ✘ |
| 12 | BRFSS: Diabetes by Age, Education, Sex, Income | 4 | ✘ |

| Task # | Task Name | $d$ | Closed Eval |
|---|---|---|---|
| 13 | BRFSS: Diabetes by Age, Education, Sex, Race, Income | 5 | ✘ |
| 14 | BRFSS: High Blood Pressure by Education, Race | 2 | ✔ |
| 15 | BRFSS: High Blood Pressure by Age, Race | 2 | ✔ |
| 16 | BRFSS: High Blood Pressure by Sex, Income | 2 | ✔ |
| 17 | BRFSS: High Blood Pressure by Race, Income | 2 | ✔ |
| 18 | BRFSS: High Blood Pressure by Age, Sex, Income | 3 | ✔ |
| 19 | BRFSS: High Blood Pressure by Sex, Race, Income | 3 | ✔ |
| 20 | BRFSS: High Blood Pressure by Age, Education, Race | 3 | ✔ |
| 21 | BRFSS: High Blood Pressure by Education, Sex, Income | 3 | ✔ |
| 22 | BRFSS: High Blood Pressure by Age, Sex, Race, Income | 4 | ✘ |
| 23 | BRFSS: High Blood Pressure by Age, Education, Sex, Income | 4 | ✘ |
| 24 | BRFSS: High Blood Pressure by Age, Education, Sex, Race | 4 | ✘ |
| 25 | BRFSS: High Blood Pressure by Age, Education, Race, Income | 4 | ✘ |
| 26 | BRFSS: High Blood Pressure by Age, Education, Sex, Race, Income | 5 | ✘ |
| 27 | BRFSS: Depression by Race, Income | 2 | ✔ |
| 28 | BRFSS: Depression by Sex, Race | 2 | ✔ |
| 29 | BRFSS: Depression by Age, Income | 2 | ✔ |
| 30 | BRFSS: Depression by Education, Race | 2 | ✔ |
| 31 | BRFSS: Depression by Age, Race, Income | 3 | ✘ |
| 32 | BRFSS: Depression by Education, Sex, Income | 3 | ✔ |
| 33 | BRFSS: Depression by Age, Sex, Income | 3 | ✔ |
| 34 | BRFSS: Depression by Sex, Race, Income | 3 | ✔ |
| 35 | BRFSS: Depression by Age, Education, Sex, Race | 4 | ✘ |
| 36 | BRFSS: Depression by Age, Education, Sex, Income | 4 | ✘ |
| 37 | BRFSS: Depression by Education, Sex, Race, Income | 4 | ✘ |
| 38 | BRFSS: Depression by Age, Sex, Race, Income | 4 | ✘ |
| 39 | BRFSS: Depression by Age, Education, Sex, Race, Income | 5 | ✘ |
| 40 | MEPS: Health Insurance by Education, Sex | 2 | ✔ |
| 41 | MEPS: Health Insurance by Age, Sex | 2 | ✔ |
| 42 | MEPS: Health Insurance by Age, Race | 2 | ✘ |
| 43 | MEPS: Health Insurance by Education, Race | 2 | ✔ |
| 44 | MEPS: Health Insurance by Age, Education | 2 | ✘ |
| 45 | MEPS: Health Insurance by Sex, Race | 2 | ✔ |
| 46 | MEPS: Health Insurance by Education, Sex, Race | 3 | ✔ |
| 47 | MEPS: Health Insurance by Age, Sex, Race | 3 | ✘ |
| 48 | MEPS: Health Insurance by Age, Education, Sex | 3 | ✘ |
| 49 | MEPS: Health Insurance by Age, Education, Race | 3 | ✘ |
| 50 | MEPS: Health Insurance by Age, Education, Sex, Race | 4 | ✘ |
| 51 | NSDUH: Cigarette Use (Last 30d) by Education, Sex | 2 | ✔ |
| 52 | NSDUH: Cigarette Use (Last 30d) by Age, Sex | 2 | ✔ |
| 53 | NSDUH: Cigarette Use (Last 30d) by Age, Race | 2 | ✔ |
| 54 | NSDUH: Cigarette Use (Last 30d) by Education, Race | 2 | ✔ |
| 55 | NSDUH: Cigarette Use (Last 30d) by Age, Education | 2 | ✔ |
| 56 | NSDUH: Cigarette Use (Last 30d) by Sex, Race | 2 | ✔ |
| 57 | NSDUH: Cigarette Use (Last 30d) by Education, Sex, Race | 3 | ✔ |
| 58 | NSDUH: Cigarette Use (Last 30d) by Age, Sex, Race | 3 | ✔ |
| 59 | NSDUH: Cigarette Use (Last 30d) by Age, Education, Sex | 3 | ✔ |
| 60 | NSDUH: Cigarette Use (Last 30d) by Age, Education, Race | 3 | ✘ |
| 61 | NSDUH: Cigarette Use (Last 30d) by Age, Education, Sex, Race | 4 | ✘ |
| 62 | NSDUH: Marijuana Use by Education, Sex | 2 | ✔ |
| 63 | NSDUH: Marijuana Use by Age, Sex | 2 | ✔ |
| 64 | NSDUH: Marijuana Use by Age, Race | 2 | ✔ |
| 65 | NSDUH: Marijuana Use by Education, Race | 2 | ✔ |
| 66 | NSDUH: Marijuana Use by Age, Education | 2 | ✔ |
| 67 | NSDUH: Marijuana Use by Sex, Race | 2 | ✔ |
| 68 | NSDUH: Marijuana Use by Age, Education, Race | 3 | ✘ |
| 69 | NSDUH: Marijuana Use by Age, Sex, Race | 3 | ✔ |
| 70 | NSDUH: Marijuana Use by Education, Sex, Race | 3 | ✔ |

| Task # | Task Name | $d$ | Closed Eval |
|--------|-----------|-----|-------------|
| 71 | NSDUH: Marijuana Use by Age, Education, Sex | 3 | ✔ |
| 72 | NSDUH: Marijuana Use by Age, Education, Sex, Race | 4 | ✘ |
| 73 | NSDUH: Cocaine Use by Education, Race | 2 | ✔ |
| 74 | NSDUH: Cocaine Use by Education, Sex | 2 | ✔ |
| 75 | NSDUH: Cocaine Use by Age, Race | 2 | ✔ |
| 76 | NSDUH: Cocaine Use by Age, Education | 2 | ✔ |
| 77 | NSDUH: Cocaine Use by Sex, Race | 2 | ✔ |
| 78 | NSDUH: Cocaine Use by Age, Sex | 2 | ✔ |
| 79 | NSDUH: Cocaine Use by Age, Sex, Race | 3 | ✔ |
| 80 | NSDUH: Cocaine Use by Age, Education, Race | 3 | ✘ |
| 81 | NSDUH: Cocaine Use by Age, Education, Sex | 3 | ✔ |
| 82 | NSDUH: Cocaine Use by Education, Sex, Race | 3 | ✔ |
| 83 | NSDUH: Cocaine Use by Age, Education, Sex, Race | 4 | ✘ |
| 84 | SCF: Home Ownership by Education, Sex | 2 | ✔ |
| 85 | SCF: Home Ownership by Age, Sex | 2 | ✔ |
| 86 | SCF: Home Ownership by Age, Race | 2 | ✔ |
| 87 | SCF: Home Ownership by Education, Race | 2 | ✔ |
| 88 | SCF: Home Ownership by Age, Education | 2 | ✔ |
| 89 | SCF: Home Ownership by Sex, Race | 2 | ✔ |
| 90 | SCF: Home Ownership by Education, Sex, Race | 3 | ✔ |
| 91 | SCF: Home Ownership by Age, Sex, Race | 3 | ✔ |
| 92 | SCF: Home Ownership by Age, Education, Sex | 3 | ✔ |
| 93 | SCF: Home Ownership by Age, Education, Race | 3 | ✘ |
| 94 | SCF: Home Ownership by Age, Education, Sex, Race | 4 | ✘ |

# B    COMPARISON OF BASE & INSTRUCTION-TUNED MODELS

In this appendix, we compare the performance of base models vs. instruction-tuned models optimized via either self-instruction (Wang et al., 2022) or supervised fine-tuning (Ouyang et al., 2022) followed by reinforcement learning with human feedback (RLHF) (Christiano et al., 2017). In the main text, we analyzed the performance of instruction-tuned models on our benchmark: Mistral 7B (Jiang et al., 2023), LLaMA3 8B (Grattafiori et al., 2024), LLaMA3 70B (Grattafiori et al., 2024), Gemma3 27B (Team Gemma et al., 2025), DeepSeek 7B (Bi et al., 2024), and Phi-4 (Abdin et al., 2024). For each of these models, we compare the model's performance against the corresponding base (pre-trained) model. The only exception is the Phi-4 model, for which a base model is not released due to safety concerns (Abdin et al., 2024).

For evaluating the models on our benchmark, we focus on the 94 tasks in the high-dimensional setting (see Sec. 2.2.1 and Tab. 2 for details). The results of comparing base vs. instruction-tuned models are shown in Fig. 5a, which again indicates poor performance across all models. We note that, for all families of models (here, a family refers to a pair of base and instruction-tuned models), instruction-tuned models perform equal or better than base models. Furthermore, a comparison of base model's vs. instruction-tuned model's scores for each task is shown in Fig. 5b. The figure indicates that for a number of tasks, instruction-tuning improves performance from a zero score to a non-zero score. Less commonly, instruction tuning makes performance worse (some instances are observed for DeepSeek and Gemma 3 model families).

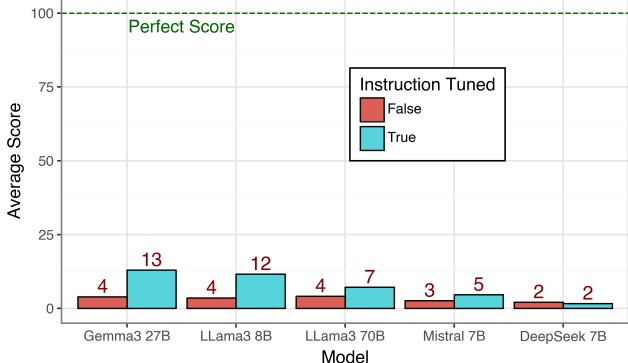

(a) Comparing the performance of base and instruct models on high-dimensional tasks.

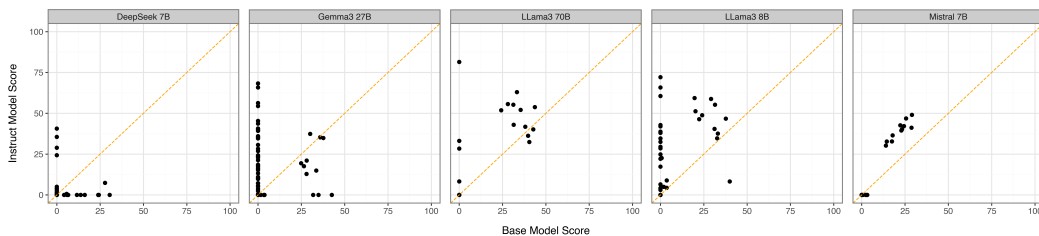

(b) Performance of base and instruct models by task and model family.

Figure 5: Performance of base vs. instruction-tuned models.

# C    CLOSED MODELS PERFORMANCE

In this appendix, we look at how the performance of closed models compares to the performance of open-weight models. In particular, we evaluate two closed models from OpenAI – GPT-4.1 (Achiam et al., 2023) and o4-mini (OpenAI, 2025), where the latter model is a reasoning language model (RLM) (Besta et al., 2025). Since these models require access to an API, we evaluate them on a subset of the tasks in the high-dimensional setting (consisting of tasks with a univariate binary $V_Y$, and $|V_X| \in \{2, 3, 4, 5\}$). In particular, we select 63 out of 94 tasks (the selected tasks are indicated in the Closed Eval column of Tab. 2), focusing on tasks for which at most 250 queries need to be evaluated on the model.

To elicit the model's distribution $\tilde{P}(V_Y \mid V_X)$, we use the likelihood-prompting technique, described in detail in Appendix E. We choose likelihood prompting, since closed models such as GPT-4.1 and o4-mini do not provide access to next-token prediction probabilities, meaning that question/answer-prompting used in the main text becomes possible only via Monte Carlo sampling. This, however, would increase the number of queries sent to the models and the associated cost by at least two orders of magnitude. This is due to the fact that even with $n_{mc} = 100$ Monte Carlo samples, the 95% confidence interval for a probability $\tilde{P}(V_Y = 1 \mid V_X = v_X) \in [0, 1]$ is only guaranteed to be reduced to a width of approximately 0.1, still reflecting a high level of uncertainty.

The performance of GPT-4.1 and o4-mini models on the 63 selected tasks is shown in Fig. 6. The

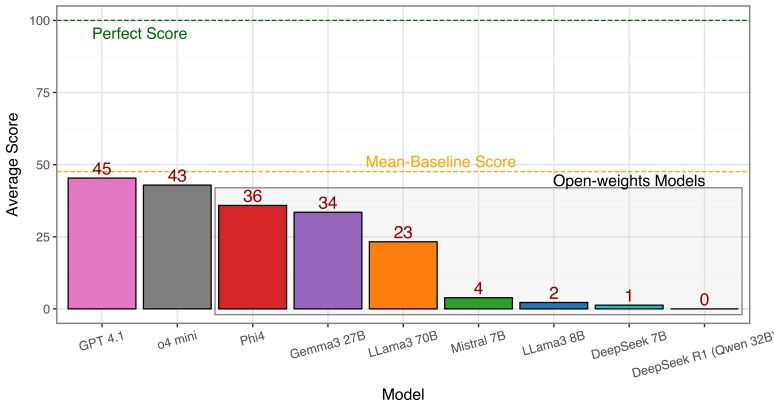

Figure 6: Performance of closed OpenAI models GPT-4.1 and o4-mini.

figure shows that GPT-4.1 and o4-mini outperform the evaluated open-weights models. This may be explained by the fact that GPT-4.1 is the largest model we evaluated, in terms of the parameter count (even though the number of parameters is not officially published). Even though OpenAI's models outperform the evaluated open-weights models, they still do not perform well. In Fig. 6, the orange line indicates the performance of the so-called *mean-baseline*, which outputs the overall population mean $P^{\text{mean}}(V_Y = 1 \mid V_X = v_X) = \mathbb{E}_P[V_Y]$ for all values $v_X \in \text{dom}(V_X)$ (and extended discussion on this baseline can be found in Appendix E.1). Therefore, none of the models are able to outperform such a baseline.

**Retrieval-Augmented Generation (RAG) (Lewis et al., 2020).**    Retrieval-augmented generation (RAG) is a technique in which an LLM is combined with an external retriever (Lewis et al., 2020). In this case, the model does not have to rely only on its internal knowledge, but relevant documents can be fetched from a collection of texts. To investigate the impact of RAG on our benchmark, we used the GPT 4.1 model with web access (the model can search the web for each query). To test how RAG affects performance, we focused on 11 tasks on which the GPT 4.1 model without web-based RAG had a score of 0 (corresponding to task indices 40, 43, 57, 62, 63, 64, 65, 66, 67, 70, and 71 in Tab. 2). Here, the expectation is that RAG would help the model performance, and would result in a higher score. However, the results for the GPT 4.1 model with web-based RAG still showed a zero score for each task. Therefore, perhaps surprisingly, for these tasks RAG did not result in improved performance. This finding warrants further investigation of how RAG may improve the performance of models on our benchmark. This investigation is left for future work.

# D  DETAILS ON RESPONSES AND SCORING

As mentioned in Sec. 2.2.1, we use a known strategy for eliciting model responses from previous works on question answering (Dominguez-Olmedo et al., 2024; Hendrycks et al., 2020; Santurkar et al., 2023). As models may exhibit ordering bias (Dominguez-Olmedo et al., 2024), we consider all of the different $|\mathrm{dom}(V_Y)|!$ permutations of labels $A, B, C, \ldots$, over the answers $v_Y \in \mathrm{dom}(V_Y)$, and average the probabilities accordingly (if the number of permutations $|\mathrm{dom}(V_Y)|! > 120$, we consider 120 random permutations). The above way of eliciting responses may be used for all models with open-source weights, which allows us to access next-token prediction probabilities. For the ground truth distribution $P$, in the low-dimensional setting we use a simple bin-counting estimator on the available dataset, whereas in the high-dimensional case we fit the distribution $P(V_Y \mid V_X)$ using `lightgbm` (Ke et al., 2017), with actual values $P(V_Y = v_Y \mid V_X = v_x)$ obtained out-of-sample, using 5-fold cross-validation.

# E  LIKELIHOOD VS. QUESTION-ANSWERING PROMPTS

In this appendix, we discuss an alternative way of eliciting model distributions $\tilde{P}(V_Y \mid V_X)$. We recall Ex. 1 from the main text, in which we were interested in marijuana usage ($V_Y$) over age groups ($V_X$). For this task, we used a prompt template

$$\pi(v_X) = \text{"Has a person aged } \{v_X\} \text{ ever used marijuana?"}, \tag{8}$$

which for different values of age $v_X$ creates different questions posed to the model. With the question, the model is also given the choice of answers yes/no, which are labeled with letters A, B. For extracting the model's probability $\tilde{P}(V_Y = 1 \mid V_X = v_X)$, we investigate the probabilities associated with letters A, B in the model's next-token prediction distribution $\tilde{P}^{\mathrm{nt}}$.

However, this approach, which we refer to as *question/answer-prompting* (QA-prompting, for short), is not the only way to elicit the model's distribution $\tilde{P}(V_Y \mid V_X)$. An alternative way of eliciting this distribution is to use a different prompting strategy, which asks for the probability directly:

$$\pi^{\mathrm{prob}}(v_X) = \text{"What is the probability that a person aged } \{v_X\} \text{ ever used marijuana?"} \tag{9}$$

This strategy, which we refer to as *likelihood-prompting*, is investigated in this appendix. To elicit a probability from the model, a range of responses is offered, namely responses in the following form:

```
A. 0%
B. 0%-5%
C. 5%-10%
...
U. 95%-100%
V. 100%
```

We then again inspect the next-token prediction probabilities in $\tilde{P}^{\mathrm{nt}}$, and choose the response with the highest probability:

$$\mathcal{A}^{\mathrm{resp}} = \underset{\mathcal{A} \in \{A,...,V\}}{\arg\max} \ \tilde{P}^{\mathrm{nt}}(\text{answer } \mathcal{A}). \tag{10}$$

The chosen response letter $\mathcal{A}^{\mathrm{resp}}$ is then mapped to either an interval, or a fixed value of 0% or 100%. For assigning the final predicted probability, for intervals, we take the midpoint. For instance, if the answer B. 0%-5% is chosen, we set the predicted probability to $0.025$, and so on. In this way, we elicit the probability $\tilde{P}(V_Y = 1 \mid V_X = v_X)$ of the model.

Model performances on the benchmark when using likelihood-prompting (compared with QA-prompting), on the 94 tasks in the high-dimensional setting with a binary one-dimensional $V_Y$, and $|V_X| \in \{2, 3, 4, 5\}$, are shown in Fig. 7. On top of the models used in the main text, we also include a version of DeepSeek R1 (Guo et al., 2025) distilled using Qwen2.5 32B (Hui et al., 2024). The results

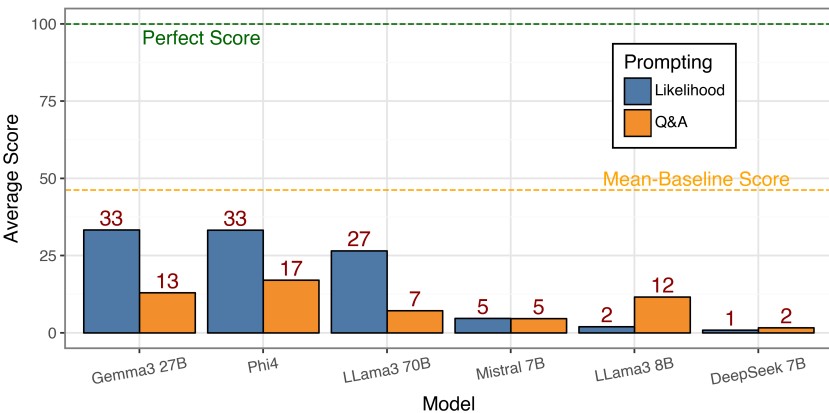

Figure 7: Models' performance with likelihood-prompting on the 94 high-dimensional tasks.

in the figure compare the performance of models on the same tasks, where likelihood-prompting is used instead of QA-prompting used in the main text. We note that the model performances, for some models, improve with likelihood-prompting (in particular, the Phi4 model moves from an average score of 17/100 to 33/100, Gemma3 27B from 13/100 to 33/100, LLama3 70B from 7/100 to 27/100). At the same time, for some models, the performance with likelihood prompting becomes worse (LLama3 8B). However, overall, model performance is still quite poor, and we further contextualize this further in the sequel.

### E.1 Baselines – how good are the models?

In the evaluation of our benchmark, we used two *baselines*, which corresponded to the notion of zero probabilistic knowledge. The first baseline was the uniform baseline $\tilde{P}^{\text{unif}}$, which is in the binary case given by:

$$P^{\text{unif}}(V_Y = 1 \mid V_X = v_X) = \frac{1}{2} \; \forall v_X. \tag{11}$$

This baseline corresponds to random guessing. In the main text, we also discussed the 0/1 baseline, which outputs either a probability 0 for events with a marginal probability $\leq 0.5$, or a probability 1 for events with a marginal probability $> 0.5$:

$$P^{0/1}(V_Y = 1 \mid V_X = v_X) = \mathbb{1}(\mathbb{E}_P[V_Y] > 0.5) \; \forall v_X, \tag{12}$$

where $\mathbb{E}_P[V_Y]$ is the true mean of $V_Y$ (in the ground truth distribution $P$). In our scoring scheme, these baselines were used to determine the score $S = 0$ for a given task (see Sec. 2.2.1). However, another interesting baseline is the *mean-baseline*, defined by:

$$P^{\text{mean}}(V_Y = 1 \mid V_X = v_X) = \mathbb{E}_P[V_Y] \; \forall v_X. \tag{13}$$

This baseline outputs a fixed value (the marginal mean of the outcome $V_Y$) for each $v_X \in \text{dom}(V_X)$. In words, this baseline is aware of the marginal probability of the event $V_Y$, but has no knowledge about the *variation of* $P(V_Y = 1 \mid V_X = v_X)$ according to $v_X$ around the marginal mean $\mathbb{E}_P[V_Y]$. This baseline corresponds to some probabilistic knowledge (marginal mean), but without any knowledge over how the probability varies across population subsets. Assigning a score of $S = 0$ to this baseline would make high scores on our benchmark even more difficult to achieve, and an argument could be made for including it in future iterations of the benchmark. In Fig. 7, the performance of the mean-baseline with respect to the current evaluation is shown (orange horizontal line), indicating an average score of 46/100. This means that the inclusion of this baseline would effectively render the scores of all models investigated in this manuscript close to 0. This observation once again confirms that the current generation of LLMs does not possess knowledge about observational distributions in the real world, regardless of the strategy (QA- or likelihood-prompting) we choose for eliciting the model distributions $\tilde{P}(V_Y \mid V_X)$.

# F  DOES FINE-TUNING IMPROVE PERFORMANCE?

In this appendix, we evaluate whether fine-tuning helps the performance of LLMs on our benchmark. Given the relatively poor performance of models across a range of tasks, a natural question arises whether additional training could improve models' knowledge of observational distributions. We investigate this hypothesis by fine-tuning a LLaMA3 8B instruction-tuned model on synthetic data generated from our ground truth datasets, and comparing the performance to the baseline pretrained model.

## F.1  SYNTHETIC DATA CREATION

Our first step is to create a synthetic dataset that simulates the type of text a model might be given access to during its pretraining stage. To generate such that, we use a pretrained Gemma3 27B instruction model (Team Gemma et al., 2025) and focus on the NSDUH data. Specifically, we sample different individuals from the dataset, reflecting the observational distribution. Upon selecting an individual, we generate a text paragraph representing the person and their associated covariates. For NSDUH data, different tasks look at cigarette, alcohol, marijuana, cocaine, and heroin usage across demographics and therefore each generated textual note must contain all of this information. In particular, we use the following prompt template for querying the Gemma3 27B model:

---

**Prompt**

```
You are a data generator. Follow the rules strictly.
RULES:
1) Write a single narrative enclosed in <story>...</story>.
2) Do NOT include headings, lists, analysis, or
any text outside the tags.
3) Mention ALL facts given below exactly once (age, sex, race,
    education, cigarette and alcohol last-month use,
    marijuana, cocaine, and heroin ever use).
4) Keep it under 200 words.

FACTS:
- age: {age}
- sex: {sex}
- race: {race}
- education: {edu}
- alcohol last month: {alc_monthly}
- cigarettes last month: {cig_monthly}
- alcohol last month: {cig_monthly}
- marijuana ever: {mj_ever}
- cocaine ever: {coc_ever}
- heroin ever: {her_ever}

OUTPUT FORMAT:
<story>
(your narrative here)
</story>
```

---

We create a dataset of 2,500 samples. An example of a generated data sample is shown below, showcasing the type of synthetic data to be used for fine-tuning:

**Example Output**

*Ethan, a 22-year-old White male, was navigating the complexities of early adulthood. He'd started some college but hadn't yet earned a degree, feeling unsure about his path. He worked part-time at a local bookstore, enjoying the quiet atmosphere and the smell of old paper. Ethan wasn't a smoker or a drinker; he hadn't used cigarettes or alcohol in the last month. He'd never experimented with marijuana, cocaine, or heroin, preferring to stay clear of substances. His friends were often surprised by this, but Ethan was content with his choices. He spent his free time sketching in his notebook, dreaming of becoming an illustrator, and occasionally volunteering at the animal shelter.*

## F.2 FINE-TUNING

By fine-tuning a decoder-only model (LLama3 8B or Mistral 7B) on the above dataset, we test whether a model can internalize observational knowledge by learning to do causal language modeling on narrative text. An alternative fine-tuning approach would be to update model weights directly on summarized statistics. However, this way of training would no longer represent embedding probabilistic knowledge (which our benchmark aims to assess) into the model, and would instead embed factual knowledge.

Standard supervised finetuning (SFT) is used for both LLama3 8B and Mistral 7B, which updates all model weights. The cross-entropy loss for next token prediction is optimized. Each model is trained for 6 epochs using a learning rate of $5 \cdot 10^{-5}$ and the AdamW optimizer with $\beta_1 = 0.9$, $\beta_2 = 0.999$. We apply a linear learning rate scheduler and monitor the loss on both training and validation splits. We save the model checkpoints that achieve the lowest evaluation loss. Loss statistics over the 6 epochs are shown in Fig. 8. As the loss behavior indicates, the fine-tuning approach reduces both

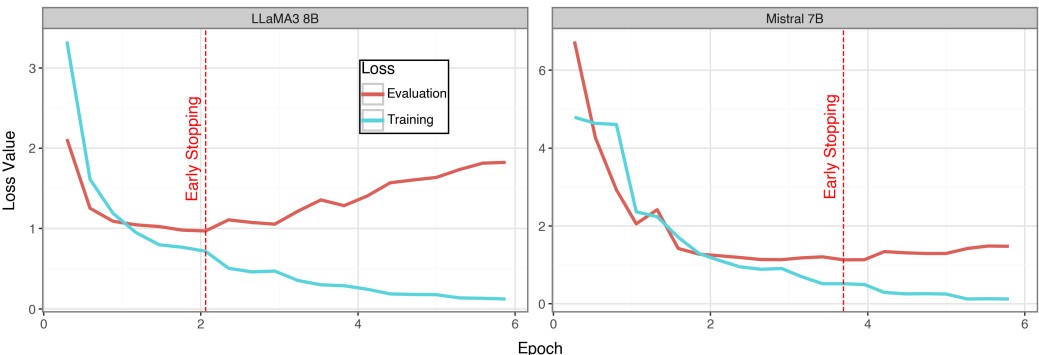

Figure 8: Training and evaluation loss for fine-tuning LLama3 8B on NSDUH-based data.

training and evaluation losses early on, after which overfitting begins, and the training loss continues to decrease while evaluation loss starts to increase. Model checkpoint saved using early stopping is indicated with a vertical red line.

## F.3 EVALUATION & OBSERVATIONS

In the last step, we evaluate the fine-tuned models on all of the NSDUH tasks, and compare the task performances to those of pretrained LLama3 8B and Mistral 7B models. The results are shown in Fig. 9 for both settings (low- and high-dimensional) and prompting techniques (question-answer and likelihood prompting). Notably, for Q&A prompting, fine-tuning does not seem to improve performance across either low- or high-dimensional settings – very few and modest score increases are recorded for both models. At the same time, in the high-dimensional setting, for some tasks where a pretrained LLama3 8B model scored above 0 the performance drops with fine-tuning.

For likelihood prompting, the impact of fine-tuning is different for low-dimensional and high-dimensional settings. In the low-dimensional setting, FT increases the performance on 6 out of 10

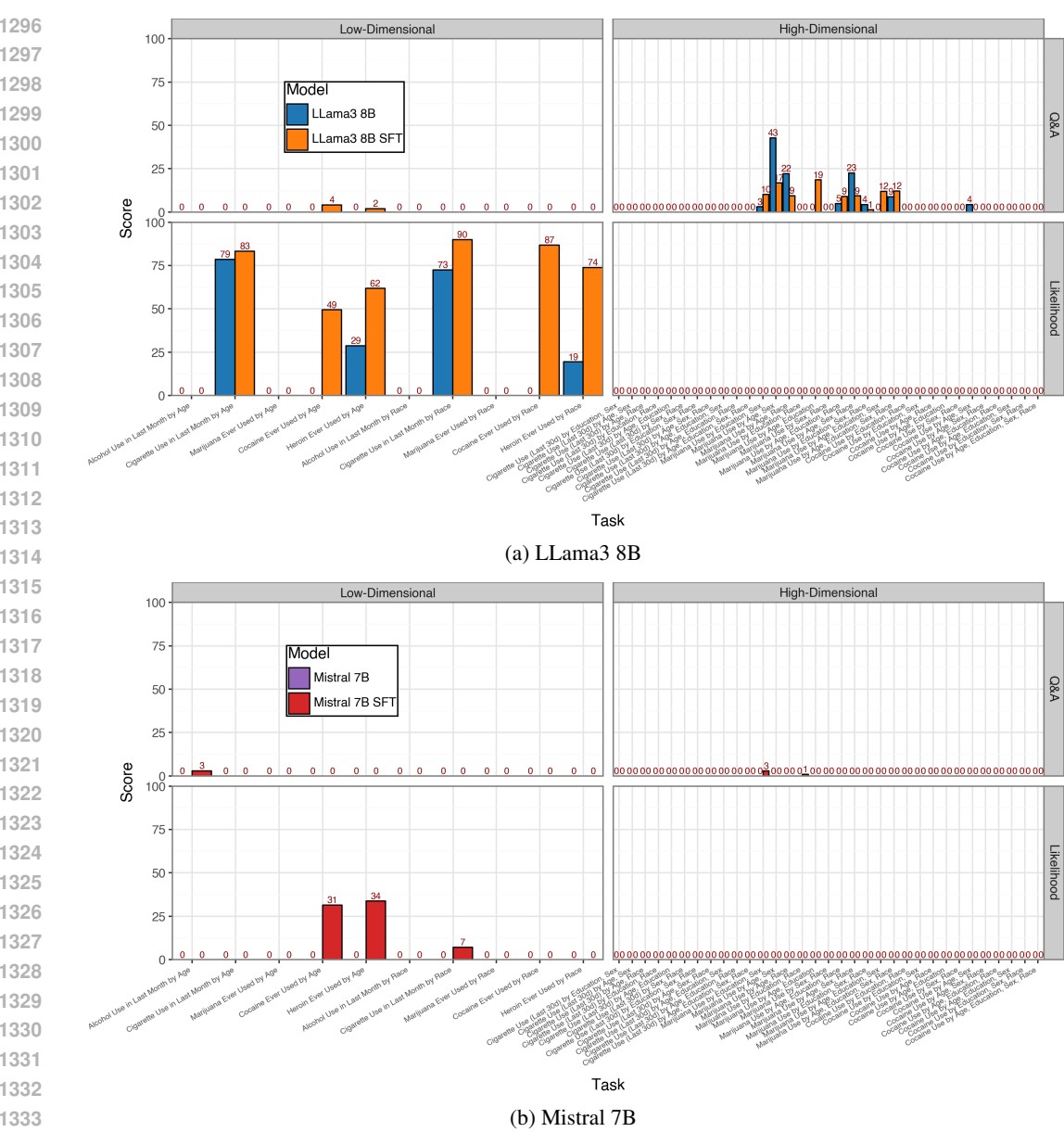

(a) LLama3 8B

(b) Mistral 7B

Figure 9: Comparison of pretrained and fine-tuned models on NSDUH-based tasks.

tasks for LLama3 8B and 3 out of 10 for Mistral 7B. In the high-dimensional setting, the performance is not improved on any of the 33 tasks for either model, so both the pretrained and fine-tuned models score exactly 0 on each task.

Therefore, some FT gains are observed for likelihood prompting in the low-dimensional setting. Interestingly, one would perhaps hypothesize that fine-tuning on the task of causal language modeling would have a greater impact and improve upon performance for the Q&A prompting, since this prompting technique more directly elicits the model's internal probability from next token predictions (instead of asking for a likelihood). However, the results do not show this. Therefore, our investigation in this appendix offers preliminary evidence showing that fine-tuning models on text generated from the correct observational distribution does not ensure that models internalize knowledge of these distributions. Future work is needed to extend these findings to other models and data sources, and to investigate methods which may improve the abilities of models in terms of probabilistic L1 knowledge.

