# OpenReview forum: "Epidemiology of Large Language Models: A Benchmark for Observational Distribution Knowledge"
_ICLR.cc/2026/Conference — ICLR 2026 Conference Withdrawn Submission_

### Official Review · Reviewer_48f4 · 2025-10-30

**Soundness:** 1
**Presentation:** 1
**Contribution:** 2
**Rating:** 2
**Confidence:** 4

**Summary:**

This paper proposes a benchmark to evaluate LLMs' inherent cognitive abilities in describing real-world population distributions, and tests both open-source and closed-source models on this benchmark.

**Strengths:**

The research motivation behind this paper is interesting. Testing LLMs' abilities in describing real-world population distributions is a good evaluation angle.

**Weaknesses:**

**Presentation:**
The presentation could be significantly improved. Sec. 2.1 and 2.2 contain excessive background details that do not directly contribute to the paper’s core message. Meanwhile, the main experimental analysis takes only about one page, which is insufficient to convey meaningful insights. The authors should better balance specificity and abstraction—focusing on the most critical aspects in the main text and moving only secondary details to the appendix.

**Soundness:**
The evaluation methodology appears questionable. For example, the paper estimates probabilities such as $P(\text{answer A})$ and $P(\text{answer B})$ from next-token likelihoods for prompts like:

> “Has a person aged 16 years ever used marijuana?
> A. Yes
> B. No”

However, this approach assumes the model will respond exactly with “A” or “B.” In practice, instruction-tuned LLMs often generate free-form text such as “According to various studies…” or “Most surveys indicate that…”. Such valid but differently structured responses are excluded from the current evaluation, which could bias the measurement and fail to capture the true underlying model belief.

**Task selection:**
Many tasks involve potentially sensitive or biased categories (e.g., employment status by sex or race). Since modern LLMs are intentionally trained to mitigate stereotypes and demographic biases, they may underperform in these tests—not because they fail to model real-world distributions, but because they are *debiased by design*. Consequently, these tasks may not accurately reflect the model’s understanding of human distributions. The benchmark would benefit from more neutral and interpretable task designs, or from prompt formulations that explicitly specify an objective statistical perspective.

**Questions:**

1. Have you examined the full next-token probability distribution for each prompt? It would be interesting to see the top 50 likely continuations to understand whether plausible alternative responses (e.g., “According to surveys…” or “Most people…”) were excluded by the current evaluation protocol. This analysis could help assess whether the current evaluation fairly captures the model’s belief distribution.

2. Many questions reflect dynamic real-world statistics that change over time. Have you considered adding temporal qualifiers (e.g., “as of 2020” or “in the past decade”) to make the evaluation more precise?

---

> ### Author Response · Authors · 2025-11-21
>
> We respond to the reviewer's comments point by point below. We note that Comment 1 is not aligned with established benchmarking practices and overlooks the references we cited.
>
> Comment 1:
> > The evaluation methodology appears questionable as the approach assumes the model will respond exactly with “A” or “B.”
>
> The actual prompt sent to the model explicitly instructs the model to answer with either A or B; Therefore, for models following instructions, this should not be a major issue.
> Furthermore, large benchmarks like MMLU, ARC, Hellaswag, PIQA, etc. require exact responses formatted A/B/C/… or similar – so our benchmark in fact follows the most-common and well-established approach in the literature – however, the reviewer calls this “questionable methodology”.
> Finally, we provided three well-known references that establish that this is the standard way to elicit model responses when investigating probabilities – which seems to have been missed or ignored by the reviewer.
>
> ---
>
> Comment 2:
> > Models may underperform not because they fail to model real-world distributions, but because they are debiased by design.
>
> Thanks, this is an interesting point that has not been covered in detail in our discussion, which we will amend in the future version.
>
> First, we note that many tasks in the benchmark do not use protected attributes such as race or gender, but instead look at education, age, and other attributes. We have run a comparison of non-sensitive tasks (not including sex/race, 51 task in total) and sensitive tasks (including sex/race, 118 tasks in total). The performance on non-sensitive tasks does not seem to be much better overall – so there is no reason to hypothesize that debiasing is the leading cause of our findings.
>
> Secondly, the point raised is something our work is also implicitly pointing to – debiasing the data may result in unintended consequences. However, investigating this aspect (fairness/alignment) requires further work and a dedicated manuscript.
>
> ---
>
> Comment 3:
> > Have you examined the full next-token probability distribution for each prompt?
>
> Thanks, we have indeed investigated different types of completions in the development process. However, such investigation raise a number of questions that are difficult to address. For instance, how does one weigh the different completions that may lead to different responses, etc? The difficulty of handling such different formatted responses is exactly why almost all benchmarks in the literature (as cited above) focus on specifically formatted answers.
>
> ---
>
> Comment 4:
> > Many questions reflect dynamic real-world statistics that change over time. Have you considered adding temporal qualifiers?
>
> We did adapt prompts to specify the year in which the dataset was collected, but we actually found that this did not change the performance meaningfully.

---

### Official Review · Reviewer_aXXc · 2025-10-30

**Soundness:** 3
**Presentation:** 3
**Contribution:** 2
**Rating:** 4
**Confidence:** 3

**Summary:**

The paper builds a benchmark to test whether LLMs encode real population distributions in the US across 10 datasets and 169 tasks. It elicits model probabilities via next token choices for multiple choice questions and scores them with an L1 distance that is normalized by baselines and calibrated with a bootstrap of the ground truth. Results show low scores overall. Best models reach about 22 out of 100 on low dimensional tasks and 17 out of 100 on higher dimensional ones. The work frames the stakes using Pearl’s Causal Hierarchy and argues that weak Layer 1 knowledge implies limits for intervention and counterfactual claims. Code and datasets are organized for reproducibility.

**Strengths:**

1. Focuses on observational distributions, not factual QA. Ties the goal to Pearl’s hierarchy with a precise statement of implications.

2. Ten public US datasets across health, education, labor, finance, crime, and attitudes. Tasks span 1d and higher conditional distributions.

3. Shows consistently poor performance across many open and closed models, and separates low vs high dimensional regimes.

**Weaknesses:**

1. Main analysis leans on multiple choice next token elicitation. Small wording or answer labeling changes can shift probabilities. The paper mentions a probability prompt variant only in the appendix.

2. High-dimensional results cover four datasets with a modest number of contexts, so evidence about the curse of dimensionality remains suggestive.

3. Some low-dimensional tasks may appear on public sites. The paper notes better scores where tables are likely online, but does not audit data leakage or pretraining overlap

4. The study isolates pure model priors, but real systems often retrieve. A RAG or table lookup baseline would contextualize how far pure priors lag.

5. Methodological novelty is limited; the main contribution is the benchmark

**Questions:**

1. How robust are scores to prompt templates, label order, and few-shot examples. It would be helpful to show a sensitivity table per task family.

2. Do calibration or temperature scaling methods change the L1 distances meaningfully, especially for models that are miscalibrated on probabilities?

---

> ### Author Response · Authors · 2025-11-21
>
> We respond to the reviewer's comments point by point below. Multiple comments seem to ignore or overlook material already presented in the manuscript.
>
> Comment 1:
> > Main analysis leans on multiple choice next token elicitation. Small wording or answer labeling changes can shift probabilities.
>
> Regarding next-token probabilities, we mentioned explicitly in text that
>
> > To mitigate ordering bias we average over different answer permutations
>
> together with providing references to earlier works that use the same approach. This seems to have been overlooked or misunderstood during the review process.
>
> ---
>
> Comment 2:
> > The paper mentions a probability prompt variant only in the appendix.
>
> Example 1 in the main text mentions likelihood-prompting, and defers it to the appendix (due to space limitations), so we are slightly confused about the comment, this part may have been overlooked.
>
> ---
>
> Comment 3:
> > Evidence about the curse of dimensionality remains suggestive.
>
> Thanks – however, we note that this is acknowledged and made explicit in the discussion at the end of the paper, so we are unsure how to improve upon this further.
>
> ---
>
> Comment 4:
> > The paper notes better scores where tables are likely online, but does not audit data leakage or pretraining overlap.
>
> Thanks, this is indeed an interesting point, and part of future work. In our current benchmark, we have no access to the training data for any of the models (even the open-weights models). Therefore, auditing for overlap becomes extremely difficult. In future work, we intend to investigate this by doing pre-training on known data and determining more closely what is / is not internalized by models during pre-training.
>
> ---
>
> Comment 5:
> > The study isolates pure model priors, but real systems often retrieve. A RAG or table lookup baseline would contextualize how far pure priors lag.
>
> As mentioned in Section 2.3, we analyze the impact of both RAG and fine-tuning in the Appendices. This is yet another aspect that seems to have been missed entirely.
>
> ---
>
> Comment 6:
> > Methodological novelty is limited; the main contribution is the benchmark
>
> We note that this submission is under the category of datasets and benchmarks – and do not see how methodological novelty is a limitation.

---

### Official Review · Reviewer_rgaT · 2025-10-31

**Soundness:** 2
**Presentation:** 3
**Contribution:** 2
**Rating:** 4
**Confidence:** 3

**Summary:**

The paper introduces a benchmark that probes whether large language models encode observational, population-level distributions rather than only "factual" recall (e.g., "Who is the President of France?"). The tasks span major public US datasets across health, education, labor, finance, and crime, posed as templated questions, eliciting conditional distributions from models. Scores are computed by a distance from ground truth using an L1 formulation with permutation-averaged answer ordering. The authors position the results using Pearl’s causal hierarchy to argue that weak "Layer-one knowledge" in LLMs undercuts claims about possible interventional or counterfactual competence within LLMs. The main empirical finding is that current models fare poorly overall in distributional knowledge, with modest advantages when the queried statistics plausibly exist verbatim on the web; the paper also explores likelihood prompting, closed-weight models, retrieval, and small finetuning, all with limited apparent upside.

**Strengths:**

The motivation is overall solid: observational knowledge is a prerequisite for meaningful causal claims, and the benchmark operationalizes that premise in a way that is modular and easy to extend. The scoring method emphasizes stability and avoids KL’s brittleness; the use of permutation averaging usefully mitigates option-ordering artifacts. The paper triangulates results across open and closed models, contrasts question-answer with likelihood prompting, and even tests retrieval and finetuning; together, these analyses lend credibility to the central negative result. The paper’s framing using the causal hierarchy is helpful (although not essential to the contribution) and well-explained for a general ICLR audience. The limitations/ethics/reproducibility statements are thorough.

**Weaknesses:**

- The conceptual contrast between “factual” and “probabilistic” knowledge is possibly overstated. Many distributional summaries are themselves "facts" about the world expressed in text, so the paper could temper claims that distributional knowledge is categorically distinct from factual knowledge and instead clarify that the difference lies in aggregation and calibration rather than a deeper ontological difference.

- The paper occasionally implies that models are claimed to “approximate real-world distributions” in general, yet the base model in LLMs is simply trained to perform next-token distributions over text. The Altman quote is suggestive, but does it set up the paper's contribution (to a degree) relative to a "strawman" argument whose prevalence in the scientific literature is not convincingly articulated? A more concrete discussion of how token likelihoods should, in the limit, generalize to world frequencies would be interesting, as would an outline of assumptions about how "facts" and "knowledge" are to be distinguished or represented in the pretraining corpus.

- The reported improvements on government statistics appear strongest where tabulations are likely present in training data; the paper notes this, but a more systematic audit of overlap or a held-out slice that cannot plausibly appear verbatim would strengthen the story about what the models do and do not internalize. If *all* distributional quantities were present in the training data, how would this affect the implications of these (or future) findings? This question is related to the overall wonder about how distributional quantities are themselves facts, or facts seen from a certain perspective, so the distinction between facts and knowledge outlined in the paper can at times be strained.

- The paper claims, "all models perform poorly"; a more objective quantification of this would be useful. For example, if humans perform far better than LLMs in this task, that would be a more defensible and objective statement than the generic "poor" performance claim, which is not defined relative to a standard that would be expected from "good" vs. "bad" performance.

- The paper's central claims are vulnerable to the possibility that LLM probabilities are well-calibrated but simply shifted. Systematic bias (but correct relative likelihoods or likelihood rankings) would I believe be quantified as lack of knowledge in this framework. Relatedly, I believe there is at least some work already indicating that LLMs might struggle to generate complex distributional quantities, while performing much better at tasks involving ranked choice, which can then yield better implied probabilities [arxiv:2306.17563; relatedly: arXiv:2403.14859].

**Questions:**

What fraction of prompts in each dataset are likely to have near-verbatim tables on the open web, and can the authors report scores conditioned on that attribute (verbatim and non-verbatim) to try to separate memorization from generalization, if this has not been done already?

**Details Of Ethics Concerns:**

None perceived.

---

> ### Author Response · Authors · 2025-11-21
>
> We thank the reviewer for reading our paper and engaging with the content. Here are some clarifications on the questions:
>
> Comment 1:
> > The conceptual contrast between “factual” and “probabilistic” knowledge is possibly overstated.
>
> We thank the reviewer for this comment – but we think there is no fundamental disagreement, it seems to be more about the wording. Indeed, population statistics could also be viewed as “facts”. Our distinction between factual and probabilistic could also be worded as logical vs. probabilistic – where logical statements can be evaluated _without uncertainty_, whereas probabilistic knowledge fundamentally requires considering uncertainty by sampling and aggregating facts over individuals or populations. In this sense, we do believe there is a genuine, fundamental distinction between the two types of knowledge – in the fact whether inherent uncertainty needs to be considered.
>
> ---
>
> Comment 2:
> > The Altman quote is suggestive, but does it set up the paper's contribution (to a degree) relative to a "strawman" argument whose prevalence in the scientific literature is not convincingly articulated?
>
> Thanks for the comment – here, we wish to point to works in the literature that are treating language models as possessing not only observational knowledge but also interventional (causal) knowledge [1]. Therefore, we believe our discussion is genuine, without attempting to have a straw man argument. Having said that, we do acknowledge the reviewer’s point – not everyone may expect these capabilities from models.
>
> ---
>
> Comment 3:
> > Audit of overlap or a held-out slice that cannot plausibly appear verbatim would strengthen the story about what the models do and do not internalize.
>
> Thanks, this is indeed an interesting point, and part of future work. In our current benchmark, we have no access to the training data for any of the models (even the open-weights models). Therefore, auditing for overlap becomes extremely difficult. In future work, we intend to investigate this by doing pre-training on known data and determining exactly what is / is not internalized by models during pre-training.
>
> ---
>
> Comment 4:
> > The paper claims, "all models perform poorly"; a more objective quantification of this would be useful. For example, if humans perform far better than LLMs in this task, that would be a more defensible and objective statement than the generic "poor" performance claim.
>
> Thanks for this comment – here, we note that previous research shows that humans are known to deviate from normative probabilistic reasoning in systematic ways  (e.g., through heuristics and cognitive biases [Kahneman and Tversky, 1972; 1974]). Therefore, we would say that AI systems should be held accountable to a higher standard than humans – given that humans may perform poorly.
>
> Regarding the second point – it is true that objectively defining “poor” is non-trivial. However, here we note that on many tasks, many of the models have a score of 0 – meaning that their beliefs are further from the truth than random guessing – which the reviewer may agree with us can be called objectively “poor”. Appendix E.1 also provides a further relevant discussion on why the model performance could be considered as poor.
>
> To account for the comment, though, we qualify our statements, stating that models cannot attain probabilistic knowledge statistically indistinguishable from the ground truth.
>
> ---
>
> Comment 5:
> > The paper's central claims are vulnerable to the possibility that LLM probabilities are well-calibrated but simply shifted.
>
> Thanks, we have indeed thought about this possibility in the development process, and have analyzed a lot of model distributions manually. From this investigation, we are pretty sure that this is not actually the case.
>
> The shared references are quite interesting, and appreciated. We will use them to help improve our discussion and make it more nuanced.
>
> [1] Piersilvio De Bartolomeis, Javier Abad, Guanbo Wang, Konstantin Donhauser, Raymond M Duch,
> Fanny Yang, and Issa J Dahabreh. Efficient randomized experiments using foundation models.
> arXiv preprint arXiv:2502.04262, 2025.

---

### Official Review · Reviewer_oU4o · 2025-11-01

**Soundness:** 3
**Presentation:** 3
**Contribution:** 2
**Rating:** 2
**Confidence:** 4

**Summary:**

The authors study the ability of LLMs to answer population-level statistical questions (e.g., what is the probability that a US citizen is male?). They establish a methodology for questioning and eliciting probabilities from open and closed weight LLMs, and assemble a benchmark of questions grounded on datasets containing data on demographics, consumer behavior, education, etc. They evaluate the models using either yes/nor question answers --  from which they elicit probabilities either by the relative frequency of repeated questioning or by renormalization of the next token probability (this only for open weight models) -- or a likelihood strategy that directly queries for the probability of some event encoded as histograms. Results show that performance is overall poor, which closed-weight models outperforming closed-weight models and likelihood questioning outperforming yes-no questioning for most models.

**Strengths:**

To my knowledge, the task investigated by this work is novel. It is certainly a useful task and questioning wether LLMs can satisfactorily answering such type of questions is scientifically and practically useful. The benchmark and methodology are reasonable. The paper is  mostly clear and well written, and cites relevant related work.

**Weaknesses:**

While the task addressed is relevant, it is not clear that those type of questions are the most relevant for the end users (say, data analysts or domain experts). The text brings no justification for the particular choice of statistical questions analysed. For example, one might be more interested in deciding the mode of a distribution or statistics (median, interquartile range etc) instead of a marginal distribution of a categorical variable. The discussion about SCMs and causality are not very relevant for this actual work; on the other hand, many important empirical results are left for appendices, as the comparison of likelihood vs. yes-no questioning, performance of closed vs. open weight and fine-tuning. This makes the work poorly self-contained.

**Questions:**

I can understand that possessing some knowledge about basic statistical facts might be beneficial for a general purpose LLM. However, one would not expect an LLM to have implicit knowledge about every dataset, even if public. So in this sense the benchmark of datasets is rather arbitrary and the results not robust. Wouldn't be better to investigate the ability of LLMs to answer such questions when given information about the dataset (say, about the data at hand or even in the form of statistics)?

---

> ### Author Response · Authors · 2025-11-21
>
> We respond to the reviewer's comments point by point below. Several of the comments are based on misunderstandings or overlooked material already presented in the manuscript.
>
> Comment 1:
> > it is not clear that those type of questions are the most relevant for the end users.
>
> Consider the following, rather plausible, real-world example: an individual experiencing symptoms of diabetes, is inquiring with a language model to learn what the probability that they have the disease is. The individual shares the information about themselves (demographics, body mass index, habits), and asks the model what the level of diabetes risk is for them. In this context, the type of knowledge investigated in this paper is not only relevant, but _essential_ for the end user. For this specific diabetes example, our benchmarking of questions on the BRFSS dataset shows that this knowledge is not available to the model. Numerous other examples can be constructed showing the importance of probabilistic knowledge for end users.
>
> Therefore, the reviewer’s comment points to a serious misunderstanding of the paper’s content.
>
> ---
>
> Comment 2:
> > The text brings no justification for the particular choice of statistical questions analysed. For example, one might be more interested in deciding the mode of a distribution or statistics instead of a marginal distribution of a categorical variable.
>
> Our benchmark analyzes various conditional distributions – it does not analyze marginal distributions only, as implied by the reviewer’s comment. Furthermore, notions of distributional distance we consider are more general than the proposed notions of statistics such as modes, medians, and so on. Therefore, our benchmark can be readily adapted to handle all of the mentioned functionals.
>
> ---
>
> Comment 3:
> > The discussion about SCMs and causality are not very relevant for this actual work.
>
> While we understand that our benchmark has implications for observational data predominantly – causal implications of the work are quite significant. In the conclusions, we shared a reference of work that uses language model for knowledge on _interventional distributions_ [1] – which is likely not plausible given our findings.
>
> [1] Piersilvio De Bartolomeis, Javier Abad, Guanbo Wang, Konstantin Donhauser, Raymond M Duch,
> Fanny Yang, and Issa J Dahabreh. Efficient randomized experiments using foundation models.
> arXiv preprint arXiv:2502.04262, 2025.
>
> ---
>
> Comment 4:
>
> > However, one would not expect an LLM to have implicit knowledge about every dataset, even if public. So in this sense the benchmark of datasets is rather arbitrary and the results not robust.
>
> We are once again surprised by the reviewer’s comment. For instance, for factual knowledge, large scale models are indeed expected to have access to correct answers for most questions – and therefore testing whether this holds for probabilistic knowledge seems like a natural direction to explore. Furthermore, works in the literature are already treating language models as possessing probabilistic knowledge [1] – making such investigations even more important. The provided reference seems to have been missed by the reviewer.
>
> Therefore, there is nothing arbitrary about the benchmark. The selection of datasets is a first step towards building a comprehensive evaluation. Moreover, our findings on model capabilities are quite similar across a range of models and datasets – showing a degree of robustness in our conclusions.
>
> ---
>
> Comment 5:
>
> > Wouldn't be better to investigate the ability of LLMs to answer such questions when given information about the dataset (say, about the data at hand or even in the form of statistics)?
>
> In Appendix B we look at the performance of fine-tuned models, which are given access to samples from the original data source. This is explicitly mentioned in Section 2.3.
>
> In Appendix C, we also look at performance with retrieval augmentation – again, not improving performance. Explicitly mentioned in Section 2.3 as well.
>
> This comment from the reviewer again indicates that key parts of the paper may have been missed during the review.

---

### Note · Authors · 2025-11-21

I have read and agree with the venue's withdrawal policy on behalf of myself and my co-authors.